# The QCD equation of state in small systems

W. A. Horowitz[1][*] and Alexander Rothkopf[2]

**1** Department of Physics, University of Cape Town, Rondebosch 7701, South Africa
**2** Faculty of Science and Technology, University of Stavanger, 4021 Stavanger, Norway

[*] wa.horowitz@uct.ac.za

## Abstract

We present first results on finite system size corrections to the equation of state, trace anomaly, and speed of sound for two model systems: 1) free, massless scalar theory and 2) quenched QCD with periodic boundary conditions (PBC). We further present work-in-progress results for quenched QCD with Dirichlet boundary conditions.

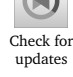
## 1 Introduction

Multiparticle correlations measurements in even the smallest collision systems are consistent with predictions from viscous relativistic hydrodynamics calculations [1]. However, these hydrodynamics calculations use a continuum extrapolated—i.e. infinite volume—equation of state. For the modest temperature probed in these small collisions, the controlling dimensionless product of the temperature and system size $T \times L \sim 400$ MeV $\times$ 2 fm / 197 MeV fm $\sim 4$ is not particularly large. One should therefore investigate the small system size corrections to the equilibrium QCD equation of state used in modern viscous hydrodynamics simulations.

## 2 Analytic Results

A natural first analytic model to test the relevance of system size on thermodynamic quantities is the massless, free scalar field put into a box, i.e. $\mathcal{L} = \frac{1}{2}(\partial_\mu \phi)^2$ with Dirichlet boundary conditions (DBCs) in 1, 2, or 3 directions of lengths $L_1$, $L_2$, and $L_3$, respectively. One may straightforwardly compute the partition function [2], then from the free energy one may compute the pressure, energy density, etc. We show in Fig. (1) the finite system size effects on the pressure as a function of temperature: the presence of the boundary limits the modes of the field, reducing the pressure from the infinite volume, Stefan-Boltzmann (SB) limit. As the dimensionless parameter $T \times L$ increases, the finite size effects decrease, as they must. As $T \times L \to 0$, the pressure converges to the $T = 0$ Casimir pressure. One can see that the finite

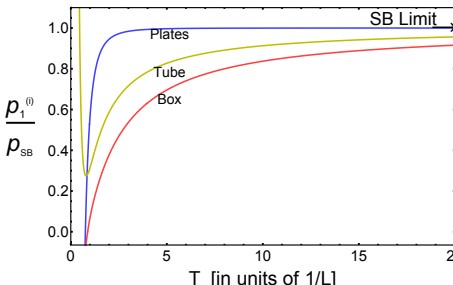
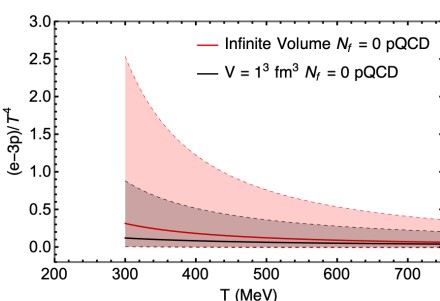

Figure 1: (Left) Finite volume pressure $p$ divided by the infinite volume, Stefan-Boltzmann limit $p_{SB}$ of a free, massless scalar field with Dirichlet boundary conditions in 1, 2, and 3 dimensions as a function of temperature $T$ in units of $1/L$, where $L$ is the length of any of the fixed, compactified directions [2]. (Right) The scaled trace anomaly $\Delta/T^4 = (\epsilon - 3p)/T^4$ as a function of temperature $T$ from $N_f = 0$ pQCD to order $g_s^5$ with scale set by $\pi T$, $2\pi T$, and $4\pi T$ in infinite volume as given in [3] and then with finite size running coupling; see text for details.

size effects are $\sim 10\%$ for even $T \times L \sim 20$ and grow as $T \times L$ decreases. It's not straight-forward to estimate $T \times L$ for hadronic collisions: after the initial nuclear overlap, the system expands longitudinally at the speed of light and transversely at half the speed of light, all while the temperature drops. For the Au and Pb nuclei collided at RHIC and LHC, the radii are $\lesssim 7$ fm. Relativistic viscous hydrodynamics suggests the initial temperature of the plasma is $\sim 400$ MeV at time $t_0 \sim 1$ fm [1]. Using the initial transverse size to roughly set $L$ we have that for central collisions, with impact parameters $b \sim 4$ fm, yield an initial $T \times L \sim 8$ and semi-central collisions, with $b \sim 9$ fm, imply $T \times L \sim 4$. Protons have a radius $\sim 1$ fm, and so for high multiplicity p+p collisions one has with $T \sim 400$ MeV that $T \times L \sim 2$. Thus with the caveats that the real systems are dynamical and nuclear, as opposed to static and scalar, the scalar field theory results suggest that the finite size effects on the thermodynamics (and thus hydrodynamics) in hadronic collisions might be significant.

One may then examine the trace anomaly $\Delta \equiv \epsilon - 3p$. The trace anomaly controls the speed of sound through $c_s^2 = \partial p/\partial \epsilon$: the larger the trace anomaly, the slower the speed of sound (and the further the speed of sound is from the conformal limit of $c_s = 1/\sqrt{3}$); the smaller the trace anomaly, the larger the speed of sound. The larger the speed of sound, the more responsive is the plasma to pressure gradients. Thus the smaller the trace anomaly, the larger the particle flow. For fixed experimentally measured flow, a smaller trace anomaly therefore implies a larger viscosity required to describe the data.

One may show that the trace anomaly in the case of a compactified massless free scalar field is identically 0, independent of any choice of conformality-breaking non-trivial finite geometry. One may further see that for a coupled $\lambda \phi^4$ theory, the trace anomaly in infinite volume is identically 0 up to fixed order $\lambda^2$; note that the dynamically generated Debye mass $m_D \sim \lambda T$ impacts the pressure and energy density at order $\lambda^{3/2}$ [5].

The trace anomaly becomes non-zero when the coupling is allowed to run and when the order of the expansion of the partition function yields logs of the temperature. Inspired by the finite size correction to the running coupling in $\lambda \phi^4$ theory [6], we took as an ansatz $g_s(\mu, L) = g_s(\mu) - \frac{3}{2} g_s^2(\mu) \frac{3}{2} \left[ (2\pi)^3 m_D(\mu)L \right]^{-1/2}$ in the $g_s^5$ QCD partition function as given in [3]. We plotted the results for the scaled trace anomaly $\Delta/T^4 \equiv (\epsilon - 3p)/T^4$ in infinite volume and in a box of sides $L = 1$ fm in Fig. (1). We conservatively estimated the uncertainty in the result by varying the scale $\mu$ from $\pi T$ to $4\pi T$. One can see that the reduction in the coupling from the infinite volume limit due to the finite system size significantly reduces the

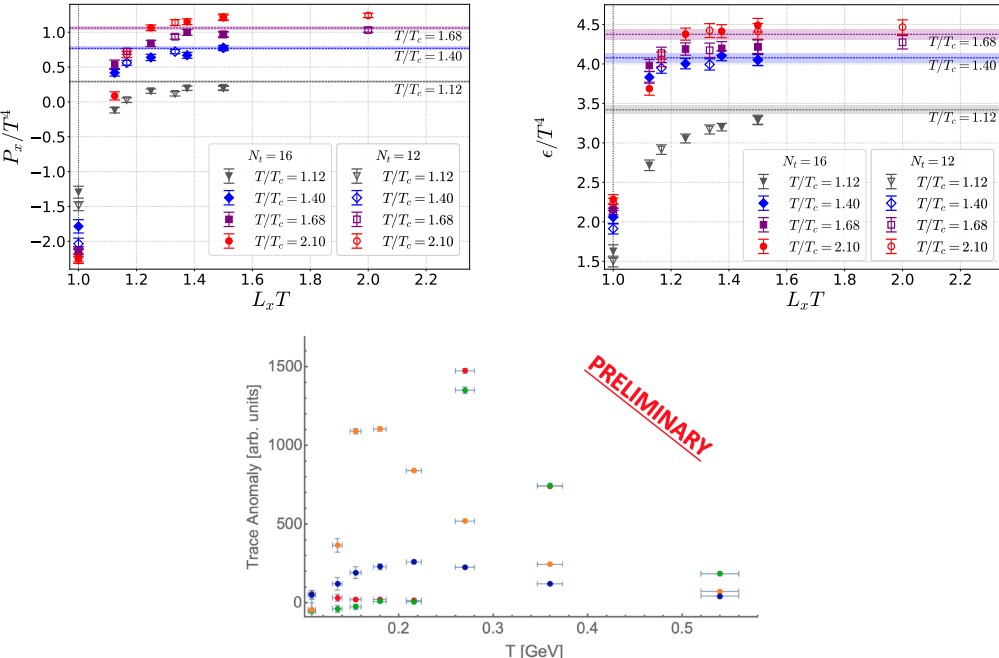

Figure 2: (Top left) Scaled pressure $p/T^4$ and (top right) energy density $\epsilon/T^4$ of quenched QCD with periodic boundary conditions with one side of fixed (small) length $L$ [7]. (Bottom) The scaled trace anomaly $\Delta/T^4 \equiv (\epsilon - 3p)/T^4$ as a function of temperature $T$ in quenched QCD with periodic boundary conditions with sides $N_x = 24$ (red),16 (green), 8 (orange) and 4 (blue) at fixed coupling [8]

.

trace anomaly, which will lead to a faster speed of sound.

## 3  Lattice QCD Results

With the intuition from our analytic calculations in hand, we may now quantify the finite system size effects on thermodynamic quantities using the lattice. Quenched QCD pure SU(3) gauge lattice simulations were performed on lattices with anisotropic spatial volumes with periodic boundary conditions (PBCs) [7]. The energy-momentum tensor defined through the gradient flow was used for the analysis of the stress tensor on the lattice. We show in Fig. (2) clear finite-size effects in the pressure and the energy density. Note that in quenched QCD $T_c \approx 270$ MeV. As anticipated from the analytic results, the pressure and energy density are reduced compared to their infinite volume limit as the dimensionless parameter $T \times L_x$ decreases.

We performed additional lattice calculations in quenched QCD with PBCs to examine the trace anomaly as a function of the temperature alone [8]. One can see in Fig. (2) that as the system size decreases, the phase transition broadens and the trace anomaly is reduced, as expected from our analytic results.

## 4  Discussion and Conclusions

We showed that for a free, massless scalar field, placing the system in a finite-sized box reduces the pressure and energy density as compared to the infinite volume limit. In p+p collisions, $L \sim 1$ fm, so $T \sim 400$ MeV implies $T \times L \sim 2$, which is far from large and we can see from Fig.

(1) that finite size corrections are $\sim 50\%$. These finite size effects persist out to surprisingly large $T \times L \sim 20$; in semi-central A+A collisions with $L \sim 10$ fm, $T \times L \sim 20$ corresponds to a QGP temperature of $\sim 400$ MeV. Thus finite size effects on the thermodynamics may have non-trivial implications for hydrodynamic modelling of heavy-ion collisions, especially in small- to moderate-sized systems. We next analytically examined the finite size effects on the $N_f = 0$ QCD trace anomaly, finding that the finite system size reduced the anomaly compared to the infinite volume limit, suggesting a faster-than-expected speed of sound. Detailed quenched QCD calculations with periodic boundary conditions agreed qualitatively with the analytic results: pressure, energy density, and the trace anomaly were all reduced compared to their infinite volume limits.

Future work includes rigorously determining the finite size effects on the running coupling in QCD, implementing Dirichlet boundary conditions (DBCs) in the lattice simulations [9], and extending the lattice simulations to full QCD. We anticipate that both DBCs and the presence of fermions will increase the magnitude of the finite size corrections seen in the lattice results presented here.

## Acknowledgements

WAH gratefully acknowledges support from the South African National Research Foundation and the SA-CERN Collaboration. AR gratefully acknowledges support from the Research Council of Norway under the FRIPRO Young Research Talent grant 286883. The numerical simulations have been partially carried out on computing resources provided by UNINETT Sigma2 – the National Infrastructure for High Performance Computing and Data Storage in Norway under project NN9578K-QCDrtX "Real-time dynamics of nuclear matter under extreme conditions."

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
