# Peer review of "The QCD Equation of State in Small Systems"

_SciPost Physics Proceedings, doi:SciPost Phys. Proc. 10, 025 (2022)_

## Round 1 · Referee Report · Anonymous (Referee 1) · 2022-2-16

Strengths

The structure in terms of a mix of analytic toy model and results from more realistic lattice calculations, with physical comparisons of the two is very nice, and mostly accessible.

Weaknesses

The abstract seems to start mid-paragraph! Some initial context appears to have been accidentally trimmed.

In section 2, should the "phenomenologically relevant T ∼ 400 MeV" be phenomenologically *irrelevant*, since the volume is large and the finite-size effects small? This seems to be the case for the "plates" picture, but not the others, which fit the ~10% effect in the text. This part could do with some expansion to explain the relevance of the plates/tube/box labels, and the logic being used to argue T and TxL values in A+A and p+p systems: to a non-specialist like myself, this bit is interesting but opaque.

It would be good to give more physics context to the trace-anomaly discussion and conclusion: what effect can this reduced coupling / increased speed of sound be expected to have on e.g. flow observables?

Report

A well-written contribution. I would just request a few improvements to the text (cf. the "weaknesses" field) to make it more comprehensible by non-specialists who can't follow the implicit logic or significance in some of the arguments made.

Requested changes

See three issues raised in "weaknesses"

---

## Round 2 · Author Response

Revised manuscript to address minor revision request from referee.

---

## Round 2 · List of Changes

The referee commented that:

1) "The abstract seems to start mid-paragraph! Some initial context appears to have been accidentally trimmed."

--Response: Yes! You're absolutely right. Thank you very much for catching this. Changed.

2) "In section 2, should the "phenomenologically relevant T ∼ 400 MeV" be phenomenologically irrelevant, since the volume is large and the finite-size effects small? This seems to be the case for the "plates" picture, but not the others, which fit the ~10% effect in the text. This part could do with some expansion to explain the relevance of the plates/tube/box labels, and the logic being used to argue T and TxL values in A+A and p+p systems: to a non-specialist like myself, this bit is interesting but opaque."

--Response: This is a nice comment, thank you. We added some sentences in the first paragraph of Section 2 expanding on the relevance for the small system size phenomenology for RHIC and LHC collisions for the more general reader

3) "It would be good to give more physics context to the trace-anomaly discussion and conclusion: what effect can this reduced coupling / increased speed of sound be expected to have on e.g. flow observables?"

--Response: This is another nice comment, thank you. We added a new second paragraph to Section 2 defining the trace anomaly in more detail and giving more insight into the phenomenological relevance of the speed of sound on flow and the extracted viscosity of the QGP fluid

---

## Editorial Decision

published